# Short-Term Outcomes of the Boston Brace 3D Program Based on SRS and SOSORT Criteria: A Retrospective Study

**DOI:** 10.3390/children9060842

**Published:** 2022-06-07

**Authors:** James H. Wynne, Lauren R. Houle

**Affiliations:** Boston Orthotics & Prosthetics, Avon, MA 02322, USA; lhoule@bostonoandp.com

**Keywords:** adolescent, scoliosis, braces, 3D bracing

## Abstract

Background: Adolescent idiopathic scoliosis (AIS) is characterized by a lateral curvature of the spine with a Cobb angle greater than 10 degrees, accompanied by rotation of the vertebral body. Bracing has been shown to be effective in halting the progression of at-risk curves, and, in some cases, even improving the Cobb angle by 6° or more. The Boston Brace 3D is part of the Boston Orthotics and Prosthetics standardized scoliosis program. The orthosis is custom-fabricated from scans, computer-aided design (CAD), and computer-aided manufactured (CAM) thoracolumbosacral orthosis used in the non-operative management of AIS. Aim: To evaluate the outcomes of a scoliosis program utilizing the Boston Brace 3D orthosis for patients with AIS, based on SRS and SOSORT criteria. Design: Retrospective study. Methods: An electronic medical records search was conducted to identify first-time brace wearers fitted between 1 January 2018, and 30 June 2019, at Boston Orthotics and Prosthetics Boston area clinics that met the SRS/SOSORT research guidelines. The initial out-of-brace, in-brace, and last follow-up X-rays (taken at least 12 months after fitting) were compared. Results: 84% of patients presenting with a single curve and 69% of patients with a double curve saw their curves improve (reduced 6° or more) or remain unchanged (±5°). Thirty-one patients started with a single curve between 25° and 30°, and thirty-two presented at 30° or below. Fifty-nine patients started with a double curve between 25° and 30°, and 59 patients presented at 30° or below. In general, the patients who wore their brace for more hours per day saw improved results. Conclusion: The Boston Brace 3D program is effective in controlling (and in some cases improving) curve progression in the non-operative management of adolescent idiopathic scoliosis. The approach is a repeatable system, as shown in this cohort of thirteen clinicians across six area clinics following the Boston Brace 3D clinical guidelines.

## 1. Introduction

Adolescent idiopathic scoliosis (AIS) is characterized by three-dimensional changes in vertebral body morphology and orientation. It is marked by a lateral curvature of the spine with a Cobb angle of greater than 10 degrees, accompanied by rotation of the vertebral bodies in the transverse plane. Studies have shown that approximately 3% of children may develop scoliosis, with only 0.3% to 0.5% progressing to curves requiring more involved intervention than observation alone. Treatment recommendations (observation, physiotherapeutic scoliosis-specific exercises, bracing (full- and night-time), surgery) and combinations of treatment are based on several factors, such as the growth remaining, curve type, and severity. The younger the patient at diagnosis, the greater the risk of progression due to the amount of growth remaining, particularly during pubertal growth spurts [1,2,3,4,5]. Bracing has been shown to be effective in decreasing the progression of at-risk curves to surgical levels, with higher average hours per day of wear time associated with improved results, and, in some cases, even improving the Cobb angle by 6° or more [5,6,7,8,9]. 

The Boston Brace 3D is part of the Boston Orthotics & Prosthetics standardized scoliosis program, containing both clinical and fabrication guidelines. The orthosis is a custom-fabricated from scans, computer-aided design (CAD), and computer-aided manufacturing (CAM) thoracolumbosacral orthosis (TLSO) used in the non-operative management of AIS since its implementation in 2018 (Figure 1).

The design of the orthosis focuses on the reduction in the patient’s scoliotic curvature(s) by addressing the spine in three planes of motion, while respecting the patient’s sagittal balance. This study retrospectively assessed the outcomes of the Boston Brace 3D program in the non-operative management of AIS based on the Scoliosis Research Society (SRS) and the International Society on Scoliosis Orthopedic and Rehabilitation Treatment (SOSORT) criteria [10].

## 2. Materials and Methods

This was a retrospective study. Internal review board (IRB) exemption was obtained for this retrospective study by wcgIRB (www.wcgIRB.com; accessed on 2 March 2022). An Electronic Medical Records (Athena Healthcare EMR) query was conducted to identify individuals fitted with a scoliosis orthosis consecutively between 1 January 2018, and 30 June 2019, at Boston Orthotics and Prosthetics Boston area clinics. Inclusion criteria were a diagnosis of AIS, first-time brace wearer, fitted with a Boston Brace 3D orthosis, age 10–17 years at time of fitting, Risser score 0–2, primary curve Cobb angle measuring 25–40 degrees, and seen in the Boston area clinics for evaluation, fitting, and follow-up. Exclusion criteria were a diagnosis of non-idiopathic diagnosis, not a first-time brace wearer, not fitted with a Boston Brace 3D orthosis, Risser score 3 or higher, primary curve measuring less than 25 degrees or greater than 40 degrees, and not having a follow-up out-of-brace X-ray at least 12 months after the initial fitting of the brace (Figure 2). 

All Boston Brace 3D orthoses are custom-fabricated from a scan of the patient (Figure 3). The scanned image (Figure 4) is imported into the CADCAM (Rodin4D) program and is modified according to the Boston Brace 3D principles that include internal pushes and shifts with opposite areas of reliefs. The dimensions of each component are determined by the clinical exam and X-ray blueprint. Internal forces are oriented to provide an anterior/medially resultant vector while maintaining the individual’s overall sagittal balance. This modification process results in an asymmetrical model of the patient, over which the brace is fabricated.

All patients had the option to have a thermal sensor (Maxim Thermochron® temperature loggers, model DS1922L/Boston O&P software) installed in their orthosis at the time of fitting, as per the Boston O&P standard of care. There was no additional cost to the parent/caretaker to have the sensor installed. The sensor was optional; therefore, we included all patients meeting the SRS/SOSORT criteria in the study cohort regardless of their decision over whether to use an adherence monitor. This was decided because not all patients that had a sensor had complete sensor data due to data not being recorded or sensor malfunction. The sensor was installed at the fitting appointment and data were downloaded at each subsequent follow-up appointment. The treating physician prescribed the recommended hours of wear with variations in prescribed hours, from 12 to 18 h per day. Only the average hours of wear, independent of the recommended hours, are reported. A distinction was made between the first follow-up (break-in wear time) and subsequent follow-up because the patient was encouraged to wean into their recommended hours per day of wear [10].

In order to assess the quality of the program, efficacy of the Boston Brace 3D, and identify areas of improvement, the following criteria were assessed: Cobb angle of each spine segment (thoracic, thoracolumbar, and lumbar) prior to bracing, initial in-brace X-ray, and final out-of-brace X-ray at minimum 12 months of brace wear. Percentage reduction of the curve in in-brace X-ray was calculated as 100 × (initial angle in degrees–in-brace angle)/(initial angle in degrees). Final percentage correction was calculated in similar manner. According to the brace program protocol and the SRS/SOSORT recommendations, the in-brace X-ray is reviewed by the team, and appropriate adjustments were made by the orthotist based on consultation with the medical team. Subsequent in-brace X-rays are typically not taken immediately after the adjustments to allow the patient to adjust to the modifications and to reduce the amount of X-ray exposure [7,8]. The modifications were not reflected in the initial in-brace X-rays reported in this study. 

Descriptive statistical analysis was performed using Microsoft Office Excel 2016 data analysis. Data are reported as means with ± standard deviations. Other data are presented as absolute values with percentages in parentheses. One-way ANOVA tests were used to assess possible differences in subgroups [10,11].

## 3. Results

The initial chart review revealed that 656 patients, aged 10–17 years, with a diagnosis of idiopathic scoliosis, were fitted with a scoliosis TLSO in the Boston Orthotics and Prosthetics Boston, MA, USA, area clinics between 1 January 2018 and 30 June 2019. The CONSORT diagram in Figure 2 shows the selection process for our final cohort of 178 patients, of which 150 were female and 28 were male. The missing data/lost-to-follow-up group consisting of 54 patients either did not have an in-brace X-ray, their last X-ray was the in-brace X-ray, or their last subsequent out-of-brace X-ray was taken less than one year from the brace fitting. A total of 24 patients did not return to see their orthotist for adjustments after they were fitted with the orthosis, and another 8 never returned after their in-brace X-ray appointment; 20 patients have been formally discharged (returned to the orthotist for final evaluation after being discharged from bracing by their physician) from bracing to date. 

Patients were divided into two groups, based on single- or double-curve presentation. Table 1 describes the single curve group, and Table 2 describes the double curves [6,10]. In the two groups, heterogeneity was seen in the curve apices, magnitudes, age at fitting, and Risser signs. The only statistical significance was seen in the variations in the Risser sign for the single curves, and the curve magnitudes for both the single and the double curves. All the other analyses showed no statistical significance. Notably, there was no statistical significance in the break-in wear time between the curve types, magnitude, or Risser sign. This was true for both the single and the double-curve types. 

The initial Cobb angle measurements were compared with the final out-of-brace Cobb angle (at least one year after the start of bracing) and were grouped as per the SRS/SOSORT guidelines for research studies: improved (Cobb angle reduced by 6° or more), unchanged (Cobb angle ±5°), and progressed (Cobb angle increased by 6° or more) [10]. A summary of the results for the single and double curves, (Table 3 and Table 4) respectively show the adherence data for the initial break-in period and two subsequent follow-up appointments.

It is interesting to note the average hours of wear time between the three groups. Those showing improvement in the curve tended to average higher wear time than those whose curves progressed. These data match with previous findings by Weinstein, Katz, and Dolan [6,7,12]. Further study is needed to better understand the significance.

Clinically significant thresholds have been identified in the literature [10] and should be analyzed. The number of patients at the start and end of treatment that presented with curves whose thresholds were >10° ≤ 30°, >30° ≤ 50° and >50°, as measured by Cobb were reviewed. (Table 5 and Table 6).

Thirty-one of fifty-one patients started with single curves between 25 and 30 degrees, with (at the time of publication) twenty-three remaining at or below 30 degrees, seven between 31 and 50 degrees, and one having progressed to surgery. Twenty of fifty-one patients started with a single curve between 31 and 40 degrees, with nine presenting at follow-up with a curve of 30 degrees or less, ten between 31 and 50 degrees, and one having surgery. Only two patients with a single curve have progressed to surgery to date. The double-curve group saw a higher percentage of patients (54% vs. 39%) initially fitted with curves greater than 30 degrees. Fifty-nine of one-hundred-and-twenty-seven patients started with a primary double curve between 25 and 30 degrees, with thirty-six remaining at or below 30 degrees, twenty-three presenting at follow-up with a primary curve between 31 and 50 degrees, and none having had surgery to date. Sixty-eight of one-hundred-and-twenty-seven patients started with a primary double curve between 31 and 40 degrees, with twenty-three presenting at follow-up with a primary curve at or below 30 degrees, thirty-six between 31 and 50 degrees, and nine greater than 50 degrees. Five of the nine patients with curves greater than 50 degrees have had surgery, and one patient presenting between 31 and 50 degrees has had surgery. Six patients initially presenting with a double curve have progressed to surgery to date.

The in-brace corrections were also reviewed. Due to the single-curve sample size, these curves were broken into three groups (improved, unchanged, and progressed), regardless of the starting Cobb angle. The single curves that progressed 6 degrees or more averaged 51 ± 22.85%, those with no change (±5 degrees) averaged 61.27 ± 27.52%, and those showing improvement averaged 66.88 ± 29.40% degrees of correction. The primary double curves were also broken into three groups (improved, unchanged, and progressed) and then by initial Cobb angle. Table 7 shows the breakdown. These data need further analysis because there are large standard deviations and ranges of correction. It must be noted, as per the program and recommendations of the SRS/SOSORT recommendations for standard protocols, that the in-brace X-ray is reviewed by the team, and appropriate adjustments are made by the orthotist based on consultation with the medical team. Subsequent in-brace X-rays are typically not taken immediately after the adjustments to allow the patient to adjust to the modifications and to reduce the amount of X-ray exposure [13,14].

## 4. Discussion

The Boston Brace 3D orthosis is part of a standardized programmatic approach for the non-operative management of idiopathic scoliosis, which consists of both clinical and fabrication guidelines. The clinical program outlines key elements for the patient evaluation, X-ray analysis (both out-of-brace and in-brace), and brace assessment and optimization throughout the episode of care. The fabrication guidelines provide a systematic and repeatable approach that is customized to each individual scan of the patient. Each orthosis is custom-fabricated from scans of the individual with scoliosis by utilizing computer-aided design and computer-aided manufacturing (CADCAM) technology. The orthosis is fabricated in a single Boston Orthotic and Prosthetic Laboratory by skilled technicians and clinicians. As has been stated in previous studies [15,16], when describing the outcomes for a particular orthosis, it is important to distinguish the name and fabrication. The Boston Brace 3D name should only be used in reference to orthoses fabricated according to the Boston Brace 3D principles and fabricated by the laboratory of origin. Other studies should state that the design is based on the principles of the Boston Brace 3D if the actual laboratory of origin did not fabricate the orthosis. Only in this way can we compare, contrast, and improve. 

The Boston Brace 3D is an enhancement of the original Boston Brace, first described by Watts and developed by Hall and Miller in the 1970s [15]. The original system utilized prefabricated modules that were customized to the patient based on their clinical presentation and radiographic blueprint [17,18]. Emans et al. showed the effectiveness of the Boston Brace in improving and halting the progression of the curve [19]. Other long-term studies showed the effectiveness of the system with patients’ quality of life matching age-controlled cohorts [19,20,21,22]. The Boston Brace System has always included a physical therapy program [17,18,23,24], consisting of individualized exercise programs for each patient. Various physiotherapeutic scoliosis-specific exercise (PSSE) programs have been validated and are available for patients utilizing a Boston Brace 3D orthosis [25]. Most patients receiving the Boston Brace 3D have at least one physical therapy session on the day of initial fitting, after which the individual with scoliosis and their family can decide whether they wish to pursue PSSE. This is not reported here. 

Similar to the original Boston Brace System, the Boston Brace 3D is a systematic approach that includes a comprehensive clinical exam of each patient and blueprint of the radiograph, with the results of each dictating the specific brace design. A scan of the patient is taken (Figure 3). The scanned image (Figure 4) is imported into the CADCAM program for custom modification, which includes specific pushes and shifts to the model.

The scan is then modified according to the Boston Brace 3D principles, which include internal pushes and shifts opposite to the areas of reliefs, the dimensions of which are determined by clinical exam and X-ray blueprint. The internal forces are oriented to provide an anterior/medially resultant vector while maintaining the individual’s overall sagittal balance. Figure 5 shows a before-and-after modification model of a patient. Figure 6 provides a top-down transverse plane view through the apex of a right thoracic curve and shows the internal shift and push applied with the opposing area of relief. This modification process results in an asymmetrical model of the patient over which the brace is fabricated.

Internal pads are added to enhance the CAD internal forces and to allow the optimization of the orthosis after the initial in-brace radiograph and subsequent growth of the individual. Each orthosis is constructed using CAD based on standardized fabrication guidelines, and a positive foam model is created via CAM. 

The Boston Brace 3D standard of care includes the use of Maxim Thermochron® temperature loggers, model DS1922L/Boston O&P software. At evaluation, the purpose of the sensor is discussed with the parents/caregivers and patients. It is emphasized that the function of the monitor is to provide an objective report showing actual brace wear time, to enable our patients to successfully achieve wear-time adherence. Although it is the standard of care to include the wear-time sensor, patients and families make the final decision as to whether to have a sensor installed. Therefore, not all of our individuals wearing the scoliosis brace had sensor data. Further research is needed to understand why some decided to utilize a wear-time sensor and others did not.

Adherence to the recommended wear schedule varied amongst all the patients. Following the recommended brace-wear schedule can be challenging for various reasons. The importance of wear time and developing a daily routine is discussed with the families, as well as staying active and involvement in extracurricular activities and other areas of interest. More research is needed to better understand the barriers to adherence. The reasons are multifaceted and beyond the scope of this retrospective review [26,27,28,29,30,31].

According to the Boston Brace 3D clinical standards, patients are typically seen three to six weeks after their initial fitting for an in-brace X-ray and follow-up with their orthopedic physician [15,17,18,19]. In an effort to reduce the number of no-show appointments, reminder phone calls, texts, and/or email messages are sent to the families in advance of their appointment. Every effort is made to reschedule any patients that do not attend the scheduled follow-up appointment. Despite these efforts, 24 patients did not return after they were fitted with their brace, another 8 patients did not return after the first follow-up appointment, and an additional 12 never attended a follow-up appointment for an out-of-brace X-ray. This information led to edits being made to our patient informational brochures to emphasize the importance of following the recommended protocol. Patients are ideally seen prior to the in-brace X-ray to review their sensor report, make any brace optimization adjustments, answer any questions, and review the follow-up protocol. After the in-brace X-ray, further optimizations are completed if required, based on review of the radiograph. The amount of in-brace correction varied, as indicted by the high standard deviations observed in our review. This variance does relate to the magnitude and flexibility of the curve, but also to brace design. Hence, it is important to review the in-brace X-ray to not only calculate the reduction in the Cobb angle, but to optimize the design and fit of the brace by noting how it is positioned on the patient, where the pads are located, and whether the areas of relief are appropriate in size and location [32].

This retrospective review only included patients who were seen in Boston Orthotics and Prosthetics Boston, MA, USA, area clinics. Some patients opted to not return to see the orthotist but continued to be seen by their orthopedic physician at Boston Children’s Hospital in Boston and Boston Children’s Hospital of Boston in Waltham, MA, as well as four additional satellite clinics. The Cobb angle, curve apex/apices, and average wear time were documented in each patient’s medical record by the treating certified orthotist. Thirteen orthotists across six Boston area clinics recorded data as per the Boston O&P scoliosis protocol. All the clinicians were trained in the principles of the Boston Brace 3D, and their outcomes were monitored as part of a continuous improvement program.

Our data matched those of other studies [6,7,12] showing the importance of the dose in successful outcomes. The patients with single curves who progressed six degrees or more (8/55) averaged 6.9 h/day of wear time during the break-in period, while with patients who showed an improvement in the curve of six degrees or more (15/55) averaged 11.4 h/day during break-in. Those whose curves stayed within +/−five degrees (32/55) averaged 10.7 h/day of wear time during break-in. 

All of the single curves that exhibited improvements of 6 degrees or more were less than or equal to 30 degrees Cobb at follow-up, compared with 7/8 of the curves that progressed 6 degrees or more were over 30 degrees Cobb, and 1/8 of the progressive curves was over 50 degrees Cobb. Of the curves that were +/−5 degrees or essentially unchanged, 19/32 were less than or equal to 30 degrees of Cobb, 13/32 were over 30 degrees of Cobb, and 0 were over 50 degrees of Cobb. 

The patients with double curves who progressed (39/127) averaged 6.9 h/day during the break-in and never averaged 11 h or more per day for the duration of this retrospective review. The patients with double curves who improved averaged 10 h/day during their break-in period and were able to achieve an average of approximately 15 h per day for the duration of this retrospective study, whereas those with curves that remained within five degrees of the initial Cobb angle averaged 11 h per day during the break-in and were able to achieve an average of approximately 13 h per day. Further study is needed regarding wear time and discussion with individuals regarding their ability to wear the orthosis for the recommended time. Most (85%) of the patients presenting with a single curve had their curve improve or not change in magnitude, whereas 69% of the patients presenting with a double curve saw their primary curve improve or not change in magnitude.

The limitations of this retrospective review include the fact that some of the patients failed to complete the bracing program by not having an in-brace or follow-up X-ray. Additionally, not all the patients had complete objective average hours of wear-time data, and the number of days between the thermal sensors varied. Future studies should be prospective with the use of an objective adherence monitor as part of the inclusion criteria, along with strict follow-up protocols, including discharge measurements and apical vertebral rotation values, both at initial evaluation and at discharge. Patient quality of life questionnaires should also be administered at initial diagnosis, discharge, and follow-up.

## 5. Conclusions

The Boston Brace 3D program, consisting of a comprehensive evaluation of the patient and a scan of the patient’s torso, which is a systematic computer-generated manufacturing process to produce a custom-fabricated orthosis, is effective in halting the progression of the scoliotic curve, and, in some cases (27% single curves; 25% primary double curve), reducing the curve by 6 degrees or more in the non-operative management of adolescent idiopathic scoliosis. The Boston Brace 3D programmatic approach is a repeatable system, as shown in this cohort of individuals with scoliosis seen by thirteen clinicians across six area clinics managing patients according to the Boston Brace 3D clinical guidelines.

## Figures and Tables

**Figure 1 children-09-00842-f001:**
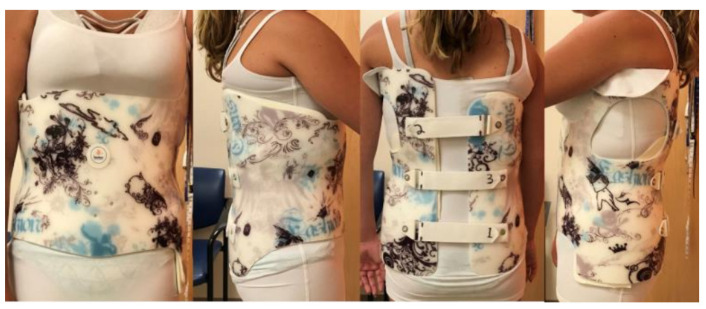
Four views of the Boston Brace 3D.

**Figure 2 children-09-00842-f002:**
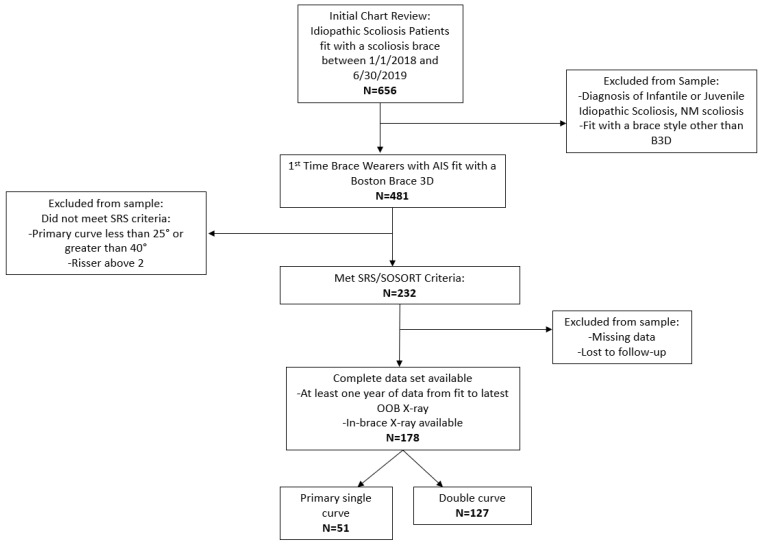
Refining of the data for final inclusion criteria.

**Figure 3 children-09-00842-f003:**
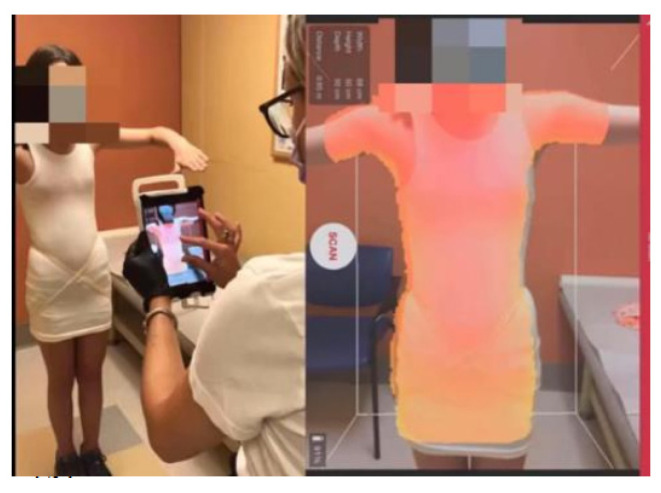
Scanning a patient for a Boston Brace 3D orthosis.

**Figure 4 children-09-00842-f004:**
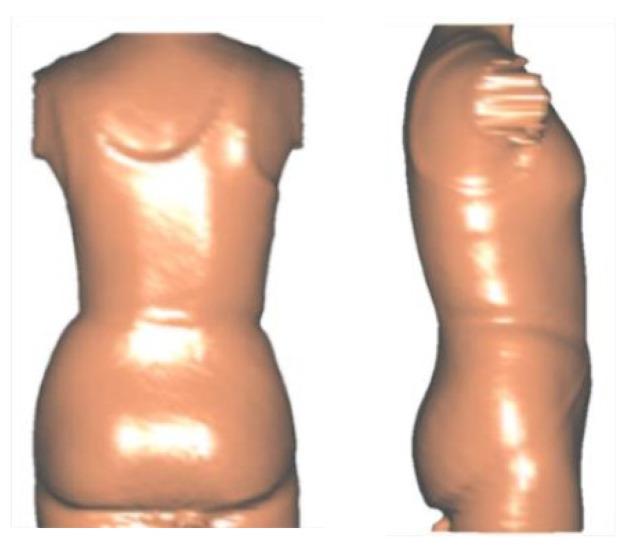
CAD model of a Boston Brace 3D orthosis.

**Figure 5 children-09-00842-f005:**
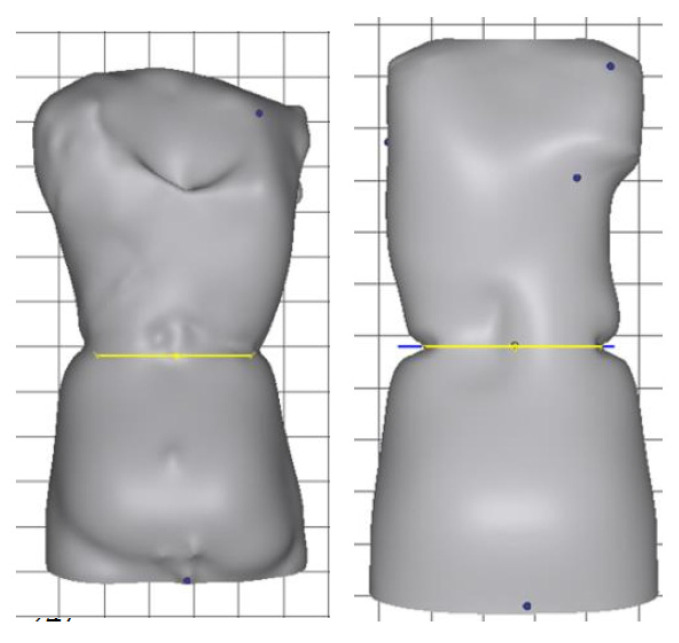
Modified scan for a Boston Brace 3D orthosis.

**Figure 6 children-09-00842-f006:**
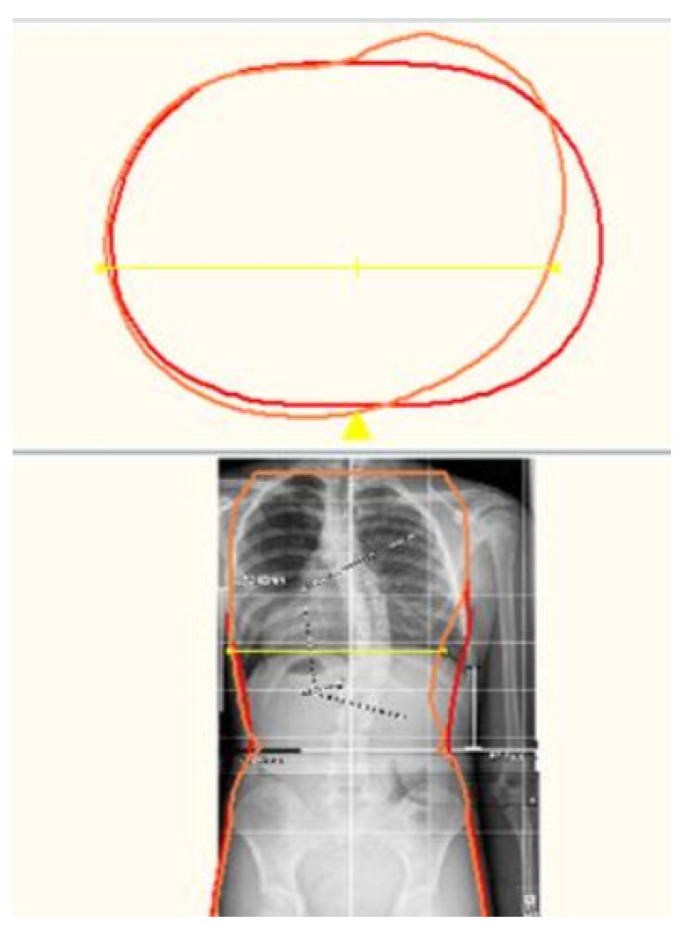
Transverse view through apex.

**Table 1 children-09-00842-t001:** Single-curve breakdown at the start of management.

Single Curves Start of Management								
Curve Apex	Number (%)	Age (Mean ± SD)	One-Way ANOVA	Female (%)	Cobb (Degrees Mean SD)	One-Way ANOVA	Average Objective Break-In Wear Time, Hours/Day (%)	One-Way ANOVA
Whole Sample	51 (100%)	13 ± 1.65		41 (80%)	30.3 ± 4.01		10.06 ± 3.83 (84%)	
Curve Type								
Thoracic	29 (57%)	11.6 ± 1.2	*p* = 0.08	23 (79%)	30.4 ± 4.1	*p* = 0.639	9.25 ± 3.67 (93%)	*p* = 0.130
Thoracolumbar	18 (35%)	13.5 ± 1.5		15 (83%)	30.6 ± 4.13		11.75 ± -3.92 (78%)	
Lumbar	4 (8%)	12.8 ± 1.21		3 (75%)	28.5 ± 3.1		9.2 ± 2.26 (50%)	
Magnitude								
25–30	31 (61%)	13.2 ± 1.52	*p* = 0.296	23 (74%)	27.5 ± 1.75	*p* = < 0.001	9.2 ± 3.81 (77%)	*p* = 0.252
31–35	14 (27%)	12.9 ± 1.94		12 (86%)	33.5 ± 1.4		11.25 ± 3.64 (100%)	
36–40	6 (12%)	12.0 ± 1.45		6 (100%)	37.5 ± 1.4		10.88 ± 4.11 (83%)	
Risser Sign								
0	20 (39%)	11.9 ± 1.4	*p* = < 0.001	17 (85%)	30.85 ± 4.3	*p* = 0.747	9.36 ± 3.8 (80%)	*p* = 0.342
1	9 (18%)	12.8 ± 1.2		7 (78%)	30.11 ± 3.14		9.16 ± 2.78 (89%)	
2	22 (43%)	14.0 ± 1.4		17 (77%)	29.9 ± 4.15		10.6 ± 3.92 (82%)	

**Table 2 children-09-00842-t002:** Double-curve breakdown at the start of management.

Double Curves Start of Management								
Curve Apex	Number (%)	Age (Mean ± SD)	One-Way ANOVA	Female (%)	Cobb (Degrees Mean SD)	One-Way ANOVA	Average Objective Break-In Wear Time, Hours/Day (%)	One-Way ANOVA
Whole Sample	127 (100%)	12.8 ± 1.3		109 (86%)	31.54 ± 4.46		9.72 ± 3.86 (79%)	
Type								
Thoracic	70 (55%)	12.64 ± 1.25	*p* = 0.265	60 (86%)	31.06 ± 4.49	*p* = 0.258	9.09 ± 3.52 (77%)	*p* = 0.200
Thoracolumbar	32 (25%)	12.97 ± 1.26		28 (88%)	32.63 ± 4.51		10.28 ± 3.83 (75%)	
Lumbar	25 (20%)	13.07 ± 1.46		21 (84%)	31.48 ± 4.22		10.66 ± 4.55 (88%)	
Magnitude								
25-30	59 (46.4%)	12.78 ± 1.10	*p* = 0.355	54 (92%)	27.61 ± 1.74	*p* = < 0.001	10.04 ± 4.03 (78%)	*p* = 0.447
31-35	37 (29.1%)	13.03 ± 1.61		29 (78%)	32.46 ± 1.30		9.95 ± 3.09 (78%)	
36-40	31 (24.4%)	12.58 ± 1.23		26 (84%)	37.90 ± 1.49		8.87 ± 4.36 (80%)	
Risser Sign								
0	67 (53%)	12.42 ± 1.16	*p* = 0.002	61 (91%)	28.34 ± 6.17	*p* = 0.538	9.41 ± 4.09 (78%)	*p* = 0.697
1	33 (26%)	13.28 ± 1.42		28 (85%)	27.09 ± 5.35		9.95 ± 3.78 (82%)	
2	27 (21%)	13.17 ± 1.22		20 (74%)	28.59 ± 1.22		10.19 ± 3.48 (78%)	

**Table 3 children-09-00842-t003:** Cobb angle changes and doses for single curves.

Cobb Angle Changes and Dose Single Curves							
Cobb Angle Change	N (%)	Break-In Average Wear Time Hours/Day	Number of Break-In Reads (%)	2nd Average Wear Time	Number of Second Reads	Third Average Wear Time	Number of Third Reads
Improved (6° or more)	14 (27%)	11.02 ± 3.63	13 (93%)	15.6 ± 3.42	13 (93%)	15.5 ± 3.62	10 (71%)
Unchanged (±5°)	29 (57%)	10.47 ± 3.79	23 (79%)	13.97 ± 4.14	27 (93%)	14.1 ± 4.13	16 (55%)
Progressed (6° or more)	8 (16%)	6.94 ± 3.07	7(88%)	10.66 ± 3.58	5 (63%)	8.2 ± 3.87	5 (63%)

**Table 4 children-09-00842-t004:** Cobb angle changes and doses for double curves.

Cobb Angle Changes and Dose Double Curves							
Cobb Angle Change	N (%)	Break-In Average Wear Time Hours/Day	Number of Break-In Reads (%)	2nd Average Wear Time	Number of Second Reads	Third Average Wear Time	Number of Third Reads
Improved (6° or more)	32 (25%)	10.2 ± 3.29	28 (88%)	14.91 ± 3.79	20 (63%)	15.6 ± 3.07	17 (53%)
Unchanged (±5°)	56 (44%)	10.98 ± 3.76	41 (73%)	12.72 ± 5.02	35 (63%)	13.29 ± 4.61	29 (52%)
Progressed (6° or more)	39 (31%)	7.6 ± 3.64	33 (85%)	7.84 ± 4.45	23 (59%)	7.67 ± 4.59	17 (44%)

**Table 5 children-09-00842-t005:** Cobb angle changes for single curves.

Primary Double Curve								
Cobb Angle at Start of Treatment				Cobb Angle at Short Term (12 Months of Treatment or More)				
Apex	Number >10° ≤ 30°	Number >30° ≤ 50°	Number >50°	Apex	Number <10°	Number >10° ≤ 30°	Number >30° ≤ 50°	Number >50°
Thoracic	37	33	0	Thoracic	0	36	40	3
Thoracolumbar	10	22	0	Thoracolumbar	1	11	12	2
Lumbar	12	13	0	Lumbar	0	11	7	4

**Table 6 children-09-00842-t006:** Cobb angle changes for double curves.

Single Curves							
Cobb Angle at Start of Treatment				Cobb Angle at Short Term (12 Months of Treatment or More)			
Apex	Number >10° ≤ 30°	Number >30° ≤ 50°	Number >50°	Apex	Number >10° ≤ 30°	Number >30° ≤ 50°	Number >50°
Thoracic	17	12	0	Thoracic	18	11	0
Thoracolumbar	11	7	0	Thoracolumbar	11	6	1
Lumbar	3	1	0	Lumbar	3	1	0

**Table 7 children-09-00842-t007:** Initial in-brace correction for double curves.

Double Curve Initial In-Brace Correction			
Cobb Angle Change	Initial Primary Degree of Cobb Angle (N)	Percent Correction (SD)	Range of Correction Percentage
Progressed (6° or more)	25–30° (18)	38.9% ± 25	6.9–100%
	31–35° (10)	30.8% ± 18.2	0–65.7%
	36–40° (11)	41.1$ ± 21.3	7.5–75%
Unchanged (±5°)	25–30° (27)	38.8% ± 24.6	0–100%
	31–35° (18)	43.2% ± 27	0–100%
	36–40° (11)	34.4% ± 16.3	5–52.6%
Improved (6° or more)	25–30° (14)	67.75% ± 26.7	24–100%
	31–35° (9)	50.45% ± 9.9	35–61%
	36–40° (9)	57.4% ± 23	17.5–100%

## Data Availability

The data presented in this study are available on request from the corresponding author.

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
