# Peer review of "Short-Term Outcomes of the Boston Brace 3D Program Based on SRS and SOSORT Criteria: A Retrospective Study"

_children, 2022, doi:10.3390/children9060842_

Round 1

Reviewer 1 Report

Dear authors,

congratulations for Your work. It seems significant and well structured, so just some minor aspects should be addressed. 

Despite it is a retrospective analysis, the research provides a great contribution to the knowledge about the scoliosis treatment. The discussion section should be improved with a short integration about other possible outcomes that might influence the adherence to the brace use. In order to do that, I suggest the following reference:

Notarnicola, A., Farì, G., Maccagnano, G., Riondino A, Covelli I, Bianchi FP, Tafuri S, Piazzolla A, Moretti, B. (2019). Teenagers’ perceptions of their scoliotic curves. an observational study of comparison between sports people and non- sports people. Muscles, Ligaments and Tendons Journal, 9(2), 225-235. doi:10.32098/mltj.02.2019.11

Then, make a correction at line 12 and 32 – 6° instead of  6⁰.

Finally, please check the editorial rules with regard to the references mention in the paper text.

Best regards and good luck

Author Response

Thank you for your feedback. The following edits have been made to the manuscript. 

  • the recommended reference has been added along with a short statement about adherence
  • Corrections made on lines 12 and 32
  • Have reviewed editorial document on refereces 

Reviewer 2 Report

There are some cocerns regarding the study. First, this is a retrospective study, with a large number of patients lost to follow up.

Specifically:

Abstract: Abstract present their study focusing mainly on the type of brace used and manufactyring process. The proceeded with conclusions without however reporting the results to support their findings.

Introduction: Authors clearly state their purpose, hence evaluate the short term outcomes of the Boston Brace 3D program in the non operative management  of adolescent idiopathic scoliosis.

Methods: Inclusion criteria adequately presented.

There is no mention of statistical methodology for analysis of results.

Results: Looking at table 1 there a large number of patients lost to follow up. Even after adjustment with inclusion criteria only 178?232 completed the study.

Lines 95 and 96 in result section are better fitted for discussion section.

Lines 181-183 in result section raises question who decides on brace modification. Is this done by the orthotist alone or based on instructions provided by the medical team. Also why there is no Xray following brace modification?

Finally, in the result section lines 172-180 standard deviations are quite high. Authors need to provide further explanation most likely in the discussion section.

Discussion: There i only one line, line 276 on the text in which authors compare their findings with other studies.

The remainder especially the part associated with describing the manufacturing process of the brace is more fit for the methods section.

Author Response

Thank you for your comments.  

  • Conclusion has been updated to reflect results
  • Statistical method mentioned. We used simple statistics in excel to calculate the mean and SD along with averaged
  • Lost to follow up discussed
  • Lines 95-96 moved to discussion area
  • Clarification on who decides on modification after the in brace x-ray and reason why subsequent x-ray not taken  provided
  • Explanation for high SD provided
  • Line 276 - reference to other studies is in regards to the importance of dose. We are just stating that our initial findings also showed a relationship, but we need to further analyze the data.
  • Brace manufacturing has been moved to the methods section. 

Reviewer 3 Report

Comment :

In this manuscript, the authors conducted a retrospective study for studying the orthosis to improve the Cobb angle in Adolescent idiopathic scoliosis (AIS). An electronic medical records search was conducted to identify first-time brace wearers fit between January, 1 2018, and June 30th, 2019 at Boston Orthotics & Prosthetics Boston area clinics that met the SRS/SOSORT research guidelines. Initial out of the brace, in the brace, and last follow-up x-rays (taken at least 12 months after fitting) were compared. Authors retrospect 656 AIS patients, find 178 cases fit with a Boston brace 3D, SRS/SOSORT criteria, and complete data set available. They found that the Boston Brace 3D program is effective in controlling curve progression in the non-operative management of adolescent idiopathic scoliosis. The work is compelling and interesting, the data is clear, and the discussions are adequate.

To make the work more profound, I would like to give three suggestions:

  • For table 2, please add the SD value to the lumbar row.
  • The row numbers overlap with tables 2, 3, and 8.
  • In tables 4 and 5, the authors show the cobb angle improves (>6 ֯) with longer average wear time hours/day, Could the authors make a histogram or table to show the cobb angle improved (>6 ֯) percentage for different average wear?

This manuscript is compelling and interesting, and the discussions are adequate. The manuscript deserves to be published.

Author Response

Thank you for your comments and  feedback. Here are the edits to the manuscript.

  • SD added to the lumbar curves section. 
  • Tables 2, 3 and 8 adjusted 
  • Regarding the histogram - there was not enough data to make this meaningful in my opinion. I this will part of future work that focus on dose. Here is was just an observation that requires further analysis. 
  •  

Round 2

Reviewer 2 Report

The authors have made some corrections to the manuscript as per previous evaluation. However two issues remain.

High ratio of patient loss in follow up

Structure of the discussion section in which a large portion is dedicated to the construction and use of the brace and also reportind the findings of the study. 

Author Response

Thank you for your comments. The following edits have been made to the manuscript.

  1. The characteristics and aim of the study have been added to the abstract. 
  2. The abstract has been divided into: Background, methods, results and conclusion. 
  3. The Canavese F. reference has been added to the bibliography and language was added on the pathogenesis of AIS.
  4. In the introduction, a brief note was added on the treatment and the Di Maria F et al reference was added. 
  5. The number of female and male participants was added to the materials e methods section.
  6. A comment was added discussing the heterogeneity in the data.
  7. Exclusion criteria was added to the materials e methods section
  8. The CONSOST diagram (Table 1) was referenced in the materials e methods section.
  9. The study design has been added to the materials e methods section
  10. Improvements for future studies has been added to the discussion section
  11. English editing services was completed, the authors are native English speakers.